# Neutrophil-to-Lymphocyte, Monocyte-to-Lymphocyte, Platelet-to-Lymphocyte Ratio and Systemic Immune-Inflammatory Index in Different States of Bipolar Disorder

**DOI:** 10.3390/brainsci12081034

**Published:** 2022-08-04

**Authors:** Katerina Dadouli, Michel B. Janho, Apostolia Hatziefthimiou, Ioanna Voulgaridi, Konstantina Piaha, Lemonia Anagnostopoulos, Panagiotis Ntellas, Varvara A. Mouchtouri, Konstantinos Bonotis, Nikolaos Christodoulou, Matthaios Speletas, Christos Hadjichristodoulou

**Affiliations:** 1Laboratory of Hygiene and Epidemiology, Faculty of Medicine, University of Thessaly, 41222 Larissa, Greece; 2Department of Psychiatry, Faculty of Medicine, University of Thessaly, 41100 Larissa, Greece; 3Laboratory of Physiology, Faculty of Medicine, University of Thessaly, 41500 Larissa, Greece; 4Primary Health Care Centre of Larisa, 41222 Larissa, Greece; 5Department of Medical Oncology, University Hospital of Ioannina, 45500 Ioannina, Greece; 6Department of Immunology and Histocompatibility, Faculty of Medicine, University of Thessaly, 41110 Larissa, Greece

**Keywords:** bipolar disorder, inflammation ratio, NLR, MLR, PLR, SII index, neutrophil, monocyte, lymphocyte, platelet

## Abstract

The neutrophil-to-lymphocyte ratio (NLR), monocyte-to-lymphocyte ratio (MLR), platelet-to-lymphocyte ratio (PLR) and systemic immune-inflammatory (SII) index, which provide a simple, rapid, inexpensive method to measure the level of inflammation, have been examined as potential inflammatory biomarkers of bipolar disorder (BD) in several studies. We conducted a case-control study recruiting 180 BD patients and 407 healthy controls. BD patients who met the inclusion criteria and were hospitalized due to BD at the psychiatry clinic of the University General Hospital of Larisa, Greece, until September 2021 were included in the study. Among them, 111 patients experienced a manic episode and 69 patients experienced a depressive episode. Data including a complete blood count were retrieved from their first admission to the hospital. Bipolar patients had a higher NLR, MLR and SII index compared to healthy controls when they were experiencing a manic episode (*p* < 0.001) and a depressive episode (*p* < 0.001). MLR was increased with large effect size only in patients expressing manic episodes. Neutrophils and NLR had the highest area under the curve with a cutoff of 4.38 and 2.15 in the ROC curve, respectively. Gender-related differences were mainly observed in the SII index, with males who were expressing manic episodes and females expressing depressive episodes having an increased index compared to healthy controls. The NLR, MLR and SII index were significantly higher in patients with BD than in healthy controls, which implies a higher grade of inflammation in BD patients.

## 1. Introduction

Bipolar disorder (BD) is a severe and chronic mental illness characterized by emotional disturbances presenting as mood phases of (hypo)mania or depression, along with changes in physical activity [1]. Worldwide, 2–3% of the overall population is affected by BD, with the mean onset at 20 years of age [2]. As one of the leading causes of disability worldwide [3], BD patients display high rates of premature mortality due to high rates of suicide, as well as high rates of comorbidity associated with cardiovascular disease, diabetes mellitus and chronic obstructive pulmonary disease (COPD) [1,4].

The pathophysiology of BD is not completely understood, although there is increasing evidence regarding the role of inflammation [5,6,7,8], as supported by a recent systematic review where the levels of interleukin-8 (IL-8), monocyte-chemoattractant protein-1 (MCP-1), eotaxin-1 and interferon-γ-induced protein 10 (IP-10), which are chemokines associated with inflammation, were found to be higher in BD patients than in healthy controls [9]. Additionally, fluctuations in serum levels of cytokines such as interleukin-1 (IL-1), IL-2, IL-4, IL-6, tumor necrosis factor (TNF)- alpha, soluble TNF receptor 1 and soluble IL-2 receptor [10,11], acute phase reactants such as high sensitivity C reactive protein (hsCRP) [12], as well as microglia activation [13] and alterations in tryptophan metabolism (via the kynurenine pathway) [14] were reported in patients with bipolar disorder.

The neutrophil-to-lymphocyte ratio (NLR), monocyte-to-lymphocyte ratio (MLR), platelet-to-lymphocyte ratio (PLR) and systemic immune-inflammation (SII) index are derived from a complete blood count, an inexpensive and reliable test that is available in most clinical settings. Higher values of these ratios indicate either neutrophilia or lymphopenia. Neutrophilia has been reported in numerous studies examining white blood counts in patients with BD [15,16] as well as in patients with unipolar depression [17], whereas the latter also present with lymphopenia. As a result, numerous studies have emerged that examine these ratios in various psychiatric illnesses among which is BD. However, opinions are divided as some studies report higher values of NLR in BD patients [18], while other studies report higher NLR and PLR [5], irrespective of current mood state. Other studies were focused on different mood states, demonstrating increased NLR [8] and PLR values [6] as well as decreased MLR values [19] in patients who experienced mania, compared to healthy controls. Studies examining mood states of BD patients also report contradictory results, with some reporting no difference between euthymic and manic phases [5], while others report significantly higher NLR, PLR and MLR in (hypo)manic than depressed individuals [20]. Overall, a meta-analysis concluded that activation of the immune system occurs during mood disorders, and NLR and PLR may be useful to detect this activation [21] 

The systemic immune inflammation index (SII) is an innovative and prognostic index, based on a combination of platelet, neutrophil, and lymphocyte counts *(SII = Plateles × Neutrophils/Lymphocytes)*. Initially, this index proved useful as a predictor of poor prognosis in solid tumors such as hepatocellular carcinoma [22], esophageal squamous cell carcinoma [23], and small cell lung cancer [24,25], as well as predicting significant coronary artery stenosis [26]. Recently the utility of this index has been extended to psychiatric illnesses. High SII index levels were associated with unipolar depression in male diabetics [27]. Another study found that SII index was higher in patients with BD during a manic phase, than either patients with BD during a depression phase or patients with major depression disorder [28]. 

The aim of this study is to compare inflammatory markers such as NLR, PLR, MLR, and SII index in patients with BD in either a state of mania or depression along with healthy controls. Moreover, we investigated potential gender differences of the aforementioned inflammatory biomarkers in bipolar disorders, and we evaluated the utility of these markers either as mood state markers or biomarkers of BD.

## 2. Materials and Methods

### 2.1. Sample

This case-control study included 180 inpatients aged from 18 to 82, diagnosed with bipolar affective disorder and hospitalized at the psychiatry clinic of the University General Hospital of Larisa, Greece, between October 2006 and September 2021. Data including a complete blood count were retrieved from their first admission to the hospital. Blood sampling was conducted before patients received their medication. All data were retrieved from each patient’s hospital record, including sociodemographic features (i.e., gender and date of birth) and clinical features (i.e., date of diagnosis, diagnosis at the time of hospitalization using DSM-IV-TR criteria, medical services provided or drugs prescribed, inpatient status, comorbid diseases, and laboratory test results). Patients were divided into two groups according to DSM-IV-TR criteria: the depressive and the manic group.

Initially, 238 patients with BD were identified while 58 were excluded from the study for various reasons: 29 had a systemic inflammatory disease (i.e., infectious diseases, chronic obstructive pulmonary disease, cardiac diseases, hematological disorders or autoimmune diseases, such as rheumatoid arthritis, inflammatory bowel disease, systemic lupus erythematosus, and gout), four were IV drug users, four patients committed suicide attended by drug overdose, one patient was pregnant and 20 patients’ medical records were unavailable. The control group consisted of 407 individuals who were age- and gender-matched with the patient group. All healthy volunteers filled out the Mood Disorder Questionnaire (MDQ) and Major Depression Inventory questionnaire (MDI) and had no personal history of any psychiatric disorders or suicide attempts [29,30]. Controls’ blood samples were taken between 7:30–9:00 a.m., thus excluding the effect of circadian rhythm on the number and type of white blood cells. The vials used for hematology blood tests were evacuated sterilized plastic or polypropylene tubes where blood is automatically aspirated, with a capacity of 3 mL. As an anticoagulant agent, EDTA Potassium Sodium (potassium Ethylene Diamanine Tetraacetic Acid) was used at a concentration of 4.55 ± 0.85 mmol/L since it prevents clotting by binding calcium. The maximum time between sampling and analysis of the samples was set at 2 h, to minimize in vitro morphological and numerical changes. Samples were analyzed at Abacus 5, Diatron of the Hematology Laboratory of Larissa Health Center with analysis ability of 26 parameters, including the five leukocyte populations, using 110 μL of whole blood. The “Abacus 5” hematology analyst combines methods to provide measurement results. It is based on the principle of volumetric impedance to determine cell concentrations and volume distributions of leukocytes, red blood cells and platelets. Optical measurement of light scattering and diffraction is used to identify the percentage of each five different leukocyte types (NEU, LYM, MONO, EOS, BASO).

Patients and controls receiving anti-inflammatory treatments (i.e., non-steroidal anti-inflammatory drugs, corticosteroids or other inflammatory drugs) or with a systemic inflammatory disease (i.e., infectious diseases, chronic obstructive pulmonary disease, cardiac diseases, hematological disorders or autoimmune diseases, such as rheumatoid arthritis, inflammatory bowel disease, systemic lupus erythematosus and gout) were excluded from the study, due to the possibility of affecting blood parameters. Patients who were pregnant, with drug intoxication or comorbid psychiatric disease, or whose blood counts during admission or hospital stay were unavailable were also excluded.

NLR, PLR, MLR and SII were calculated using neutrophil, lymphocyte, platelet and monocyte counts. The NLR, MLR, PLR and SII index were calculated using the following formulas: NLR = neutrophils count/lymphocytes count
PLR = platelets count/lymphocytes count
MLR = monocytes count/lymphocytes count
SII index = (platelets count × neutrophils count)/lymphocytes count

### 2.2. Statistical Analysis

Data were analyzed using IBM SPSS Statistics for Windows, Version.26.0 (IBM Corp., Armonk, NY, USA). Continuous variables were expressed as means ± standard deviations, and categorical variables as frequencies and percentages. Categorical data were analyzed with the use of Chi-square test. Continuous variables were checked for deviation from normal distribution (Kolmogorov–Smirnoff normality test) and for violation of assumption of homogeneity of variance (Levene’s test) for each comparison. Student’s *t*-Test and Welch test were performed for continuous data as appropriate. More precisely, when Levene’s test did not indicate a violation of homogeneity of variance (*p* > 0.05) the Student’s *t*-Test was considered appropriate. On the contrary, when a violation of homogeneity of variance was indicated (*p* < 0.05), the Welch test was used. Effect sizes for the *t*-Test (Cohen’s d) were calculated and interpreted as follows: 0.2 was considered a small effect size, 0.5 a medium effect size, and 0.8 a large effect size. All tests were 2-sided and a *p*-value of <0.05 was considered to indicate statistical significance. 

Receiver operating characteristic (ROC) curve analysis was used to determine the optimum cut-off levels of white blood cells, platelets and inflammatory ratios. Using the ROC curve, the responsiveness is described in terms of sensitivity and specificity. Values for sensitivity and for false-positive rates (1–specificity) are plotted on the y- and the x-axes of the curve and the area under the curve represents the probability a measure correctly classifies participants as patients or controls.

## 3. Results

In this study 180 patients and 409 healthy controls were included. The mean age of patients in the control group was 45 ± 13 years, 44 ± 13 years in the mania and 47 ± 11 years in the depressive phase. A total of 96 patients (53.3%) and 199 controls (48.7%) were male (Table 1).

### 3.1. White Blood Cells and Inflammatory Ratios: Differences between Depressive Episode, Manic Episode, and Healthy Control

Platelet number did not differ significantly between patients during a manic or depressive episode compared to healthy controls (Table 2). Regarding white blood cells, patients compared to healthy controls had a higher number of neutrophils during a manic or depressive episode, and higher monocyte count than healthy controls only during a manic episode (Table 2).

We then estimated the inflammatory ratios, NLR, MLR, PLR and SII index in control and patient groups. The distribution of the three inflammatory ratios of interest (NLR, PLR, and MLR) and of SII index in BD patients and healthy controls are presented in Figure 1.

BD patients had a higher NLR, MLR and SII index compared to healthy controls when they were experiencing a manic or a depressive episode, with no significant difference in PLR value (Table 3). Furthermore, non-statistically significant differences between the patients’ subgroups were found (Table 3).

The effect size was fairly large for NLR (Cohen’s d = 0.69), small-to-medium for MLR (Cohen’s d = 0.32) and medium for SII index (Cohen’s d = 0.53) in comparison between cases in a manic episode and healthy controls. Similarly, in comparison between cases in a depressive episode and healthy controls, the effect size was large for NLR (Cohen’s d = 0.94), small-to-medium for MLR (Cohen’s d = 0.26) and fairly large for SII index (Cohen’s d = 0.67).

### 3.2. White Blood Cells and Inflammatory Ratios: Differences between Depressive Episode, Manic Episode and Healthy Control in Each Sex

Compared to healthy controls, neutrophil count was increased in both male and female patients expressing a manic or depressive episode (Table 4). On the other hand, monocyte number was increased only in patients expressing a manic episode regardless of gender (Table 4). Lymphocyte number did not differ significantly in BD (male or female) patients experiencing either a manic or depressive episode (Table 4). Male patients expressing a depressive episode had decreased platelet count compared to controls, and a significant difference was found between this patients’ subgroup compared to patients expressing a manic episode (Table 4, *p* = 0.007, Cohen’s d = −0.59).

The determination of gender-related inflammatory ratios (Table 5) revealed that compared to healthy controls, NLR was increased in patients expressing a manic or depressive episode, regardless of gender with large effect size (Cohen’s d = 0.79/0.83 respectively, Table 5). Conversely, MLR was marginally increased only in male patients with manic episodes, while PLR did not have significant differences between BD patients (male or female) and healthy controls (Table 5). The SII index was increased with large effect size in male patients expressing a manic episode (Cohen’s d = 0.78) and female patients expressing a depressive episode (Cohen’s d = 0.93). A marginal increase of SII index was also found in male patients expressing a depressive episode (Table 5).

### 3.3. Predictive Ability of White Blood Cells, Platelets and Inflammatory Ratios

ROC analyses also demonstrated that neutrophils of 4.38 or above could predict the BD with 66.1% sensitivity and 71.4% specificity (area under curve (AUC) = 0.731; 95% CI: 0.685–0.777), monocytes of 0.59 or above could predict the BD with 53.9% sensitivity and 64.8% specificity (area under curve (AUC) = 0.604; 95% CI: 0.555–0.653), NLR values of 2.15 or above could predict the BD with 57.8% sensitivity and 75.3% specificity (AUC = 0.690; 95% CI: 0.643–0.737), MLR values of 0.27 or above could predict the BD with 50.6% sensitivity and 63.6% specificity (AUC = 0.582; 95% CI: 0.533–0.632) and an SII index of 516.33 or above could predict the BD with 54.4% sensitivity and 69.4% specificity (AUC = 0.652; 95% CI: 0.604–0.701). ROC analysis of area under the curve and the 95% confidence interval, cut-off levels, sensitivity, specificity, positive predictive value (PPV) and negative predictive value (NPV) are given in Table 6, and ROC curves are presented in Figure 2.

ROC analysis was also used to determine the optimum cut-off levels of white blood cells, platelets and inflammatory ratios between BD patients experienced mania and depression episode (Appendix A).

## 4. Discussion

In the present study we evaluated neutrophil, monocyte, lymphocyte and platelet counts in patients expressing a manic or depressive episode, and their gender-related differences. According to our results, neutrophil and monocyte counts were affected in BD. Specifically, all patients with either a manic or depressive episode, regardless of gender, had increased neutrophils, while monocytes were increased only in patients with a manic episode. On the other hand, lymphocytes did not differ significantly between healthy controls and patients. These findings suggest that patients with BD show changes in the innate immune system, rather than in acquired immunity.

Neutrophilia has already been described in mood disorders [31,32]. The increased neutrophil number in patients compared to healthy controls could be attributed to low-grade inflammation or/and psychological stress, as both factors may be implicated in the pathophysiology of BD. The prevalence of low-grade inflammation ranges from 22% to 40.4% in patients with BD in Western countries [33,34]. Conversely, acute and chronic psychological stress appear to impact the immune system differently. Namely, chronic stress can suppress various aspects of immunity, whereas acute stress appears to be immune-enhancing [35]. Several studies showed that stress alters neutrophil function, specifically phagocytic ability and superoxide production [36,37,38], and these alterations may be associated with depressive symptoms [39]. In our study, neutrophils were increased in patients with either a manic or depressive episode regardless of gender, compared to healthy controls. This finding is further strengthened by our ROC analysis, where the increase in neutrophils can differentiate between patients with BD and healthy controls.

Elevated monocyte counts [40] and alterations in their functionality have been observed in BD [41]. According to our results, only patients suffering from a manic episode, regardless of gender, had an increased monocyte number. Although differences in monocyte number related to the mood state have not been reported yet, differences in inflammatory response related to monocytes across mood states [42] and stages of BD illness [43] have been reported. 

In previous studies, the lymphocyte number has been found to be reduced or similar in BD patients compared to healthy subjects [17,44,45]. Our results showed no differences in the number of lymphocytes in BD patients compared to controls, but this finding could not exclude possible alterations in their immune profile, as our results were based only on complete blood count assays. 

We then evaluated the inflammatory ratios NLR, PLR, MLR and SII index. NLR was increased in all patients with a manic or depressive episode, regardless of gender compared to healthy controls. This increase is further reinforced by the ROC analysis where a 2.15 cut-off of NLR can differentiate between patients with BD and controls. Increased NLR in patients with affective disorders is a consistent and repetitive finding in many studies worldwide [21,46]. NLR appears to be higher during a manic episode and is decreased during the remission period following treatment of manic BD episodes [47,48], and thus it may predict the manic state of BD [8]. However, no significant differences in NLR between patients with manic and depressive episodes were observed in this study. Furthermore, gender differences have been reported in other studies of (hypo)manic patients [49], whereas according to our results, NLR is not affected by gender, as it was increased in both male and female patients expressing either mood episode.

According to our results, MLR is increased in patients with both manic and depressive episodes; however, there seems to be a gender and episode specific effect, as MLR is marginally increased only in males suffering from a manic episode. These results are consistent with results from other studies that reported increased MLR in patients with manic episodes [21]; in fact, increased MLR values may persist during remission compared to the control group, supporting this ratio’s utility as a trait marker of BD [48]. Furthermore, marginally significant gender differences in MLR have been reported in (hypo)manic patients [49]. It is worth noting that findings on MLR and BD state vary between studies [8,20,21,48]. Therefore, more research is required to evaluate MLR in different states of bipolar disorder, as well as to establish its utility as a trait marker of BD.

PLR is a relatively novel marker that gives information about inflammation and thrombosis and has been widely used as a prognostic indicator in cardiovascular diseases [50]. Although platelet number was decreased in male patients expressing manic episodes compared to healthy controls and patients expressing depressive episode, we found no significant differences in PLR among these groups. However, several studies present contradictory results regarding PLR. Namely, a meta-analysis reported a higher PLR value in BD patients than in healthy controls [21], while other studies showed a lower PLR value in BD patients with mixed episodes compared to healthy controls [47], or no differences in PLR between patients with BD and healthy controls [8]. These discrepancies may be attributed to differences in platelet or lymphocyte counts. For example, Özdin and Usta (2021) reported a higher PLR during the manic episodes of BD compared to the control group, but these patients also had increased platelet counts [48].

The systemic immune-inflammation index (SII) is an integrated and novel inflammatory biomarker considered to accurately reflect inflammation status. It has been widely studied in cancer [51,52], but there are a few studies in affective disorders [28,50]. We found that SII is mainly increased in males expressing manic episodes and in females expressing depressive episodes. Furthermore, ROC analysis showed that a 516.33 cut-off can be used to differentiate between patients with BD and healthy controls. Similar results are presented in previous studies where SII index was higher in BD patients suffering from a depression episode than in patients with unipolar depression [28], while another study reported a higher SII index in BD patients during a manic episode than in patients experiencing depressive episodes [53].

This study has some limitations. Firstly, patients were classified according to their mood episodes without taking into consideration the severity of their symptoms. Secondly, important variables related to the immune system, such as cytokines and CRP, were not included in our analyses. Finally, cases’ and controls’ blood samples were not taken between the same time frame (7.30–9.00 a.m.), as cases’ blood samples were taken upon admission to the psychiatry clinic. Thus, the different time of blood collection could possibly affect the results.

## 5. Conclusions

In conclusion, the NLR and the SII index are increased, regardless of gender, and do not appear to differentiate between mood states of bipolar disorder. Only the MLR, as well as monocyte count, appears to be increased with a large effect size only in patients expressing manic episodes. Gender-related differences were mainly observed in the SII index; males expressing manic and females expressing depressive episodes had an increased index compared to healthy controls. These findings contribute to the growing body of evidence suggesting that low-grade inflammation may play an important role in the pathophysiology of BD.

## Figures and Tables

**Figure 1 brainsci-12-01034-f001:**
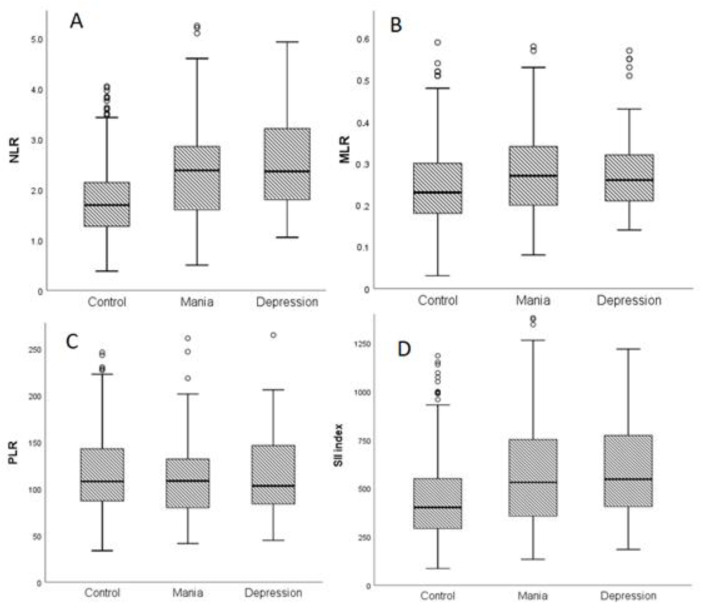
Distribution of inflammatory ratios, (**A**) neutrophil-to-lymphocyte ratio (NLR), (**B**) monocyte-to-lymphocyte ratio (MLR), (**C**) platelet-to-lymphocyte ratio (PLR) and (**D**) systemic immune-inflammatory (SII) index during a manic or depressive episode and healthy controls. (**o:** outlier).

**Figure 2 brainsci-12-01034-f002:**
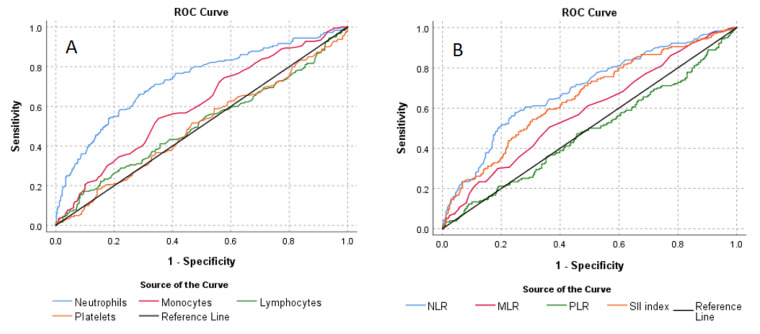
ROC curves of the of white blood cells, platelets (**A**) and Inflammatory Ratios (**B**).

**Table 1 brainsci-12-01034-t001:** Sex and age distribution of the participants per diagnosis.

		Diagnosis			Comparison between Patients and Controls
Participants		Control (*n* = 409)	Mania (*n* = 111)	Depression (*n* = 69)	
Sex	Male	199 (48.7%)	66 (59.5%)	30 (43.5%)	Pearson Chi-Square = 1.094, *p* = 0.296
	Female	210 (51.3%)	45 (40.5%)	39 (56.5%)	
Age (years ± SD)		45 ± 13	44 ± 13	47 ± 11	*t* = 0.802, *p* = 0.423

**Table 2 brainsci-12-01034-t002:** White blood cells and platelets distribution per diagnosis.

	Counts (Cells × /μL)							
	Neutrophil		Monocyte		Lymphocyte		Platelet	
	Mean ± SD	Min–Max	Mean ± SD	Min–Max	Mean ± SD	Min-Max	Mean ± SD	Min–Max
Control (*n* = 409)	3.9 ± 1.3	1.0–9.8	0.5 ± 0.2	0.1–1.2	2.3 ± 0.7	0.9–4.7	248.2 ± 58.5	127.0–477.0
Mania (*n* = 111)	5.3 ± 2.0	1.4–9.8	0.6 ± 0.2	0.2–1.2	2.4 ± 0.8	0.8–4.7	248.8 ± 61.8	100.0–470.0
Depression (*n* = 69)	5.3 ± 1.8	2.1–9.8	0.6 ± 0.2	0.3–0.9	2.2 ± 0.7	1.0–4.8	242.6 ± 68.5	100.0–498.0
Comparison between groups	*t*-value	*p*-value	*t*-value	*p*-value	*t*-value	*p*-value	*t*-value	*p*-value
Mania vs. Control	7.161	<0.001 ^w^	4.395	<0.001 ^t^	1.322	0.188 ^w^	0.095	0.924 ^t^
Depression vs. Control	6.288	<0.001 ^w^	1.613	0.107 ^t^	−0.839	0.402 ^t^	−0.714	0.476 ^t^
Mania vs. Depression	−0.027	0.978 ^t^	−1.744	0.083 ^t^	−1.579	0.116 ^t^	0.625	0.532 ^t^

^w^: welch test; ^t^: *t*-test.

**Table 3 brainsci-12-01034-t003:** Inflammatory ratios: neutrophil-to-lymphocyte (NLR), monocyte-to-lymphocyte (MLR), platelet-to-lymphocyte (PLR) and systemic immune-inflammatory (SII) index in BD patients and healthy controls of the study. Values are presented as mean ± SD.

		Mean ± SD Counts (Cells × /μL)			
		NLR	MLR	PLR	SII Index
Control (*n* = 409)		1.8 ± 0.8	0.2 ± 0.1	118.4 ± 44.7	460.3 ± 245.9
Mania (*n* = 111)		2.4 ± 1.1	0.3 ± 0.1	116.4 ± 53.3	600.4 ± 318.7
Depression (*n* = 69)		2.6 ± 1.1	0.3 ± 0.1	120.3 ± 52.5	635.3 ± 345.1
		Comparisons between groups			
Mania vs. Control	*t*-value	5.144	2.969	−0.395	4.298
	*p*-value	<0.001 ^w^	0.003 ^t^	0.693 ^t^	<0.001 ^w^
	Cohen’s d	0.69	0.32	−0.04	0.53
Depression vs. Control	*t*-value	5.305	2.002	0.320	4.043
	*p*-value	<0.001 ^w^	0.046 ^t^	0.749 ^t^	<0.001 ^w^
	Cohen’s d	0.94	0.26	0.04	0.67
Mania vs. Depression	*t*-value	−1.023	0.407	−0.478	−0.692
	*p*-value	0.308 ^t^	0.684 ^t^	0.633 ^t^	0.490 ^t^
	Cohen’s d	−0.18	0.20	−0.07	−0.11

^w^: welch test; ^t^: *t*-test.

**Table 4 brainsci-12-01034-t004:** White blood cells and platelets in male and female patients with mania or depression and healthy controls. Values are presented as mean ± SD.

		Mean ± SD Counts (Cells × /μL)							
		Male				Female			
		Neutrophil	Monocyte	Lymphocyte	Platelet	Neutrophil	Monocyte	Lymphocyte	Platelet
Control		4.0 ± 1.4	0.6 ± 0.2	2.4 ± 0.8	235.4 ± 53.3	3.8 ± 1.3	0.5 ± 0.2	2.2 ± 0.6	260.3 ± 60.6
Mania		5.7 ± 1.9	0.7 ± 0.2	2.5 ± 0.9	244.1 ± 56.3	4.8 ± 2.0	0.6 ± 0.2	2.3 ± 0.8	255.6 ± 69.3
Depression		5.2 ± 1.7	0.6 ± 0.2	2.3 ± 0.7	211.7 ± 47.2	5.3 ± 1.9	0.5 ± 0.2	2.2 ± 0.7	266.4 ± 73.2
		Comparisons between groups							
Mania vs. Control	*t*-value	6.702	2.931	0.679	1.137	3.174	2.356	1.057	0.458
	*p*-value	<0.001 ^w^	0.004 ^t^	0.498 ^t^	0.257 ^t^	0.003 ^w^	0.007 ^t^	0.369 ^w^	0.647 ^t^
	Cohen’s d	1.10	0.50	0.12	0.16	0.67	0.50	0.15	−0.07
Depression vs. Control	*t*-value	4.418	0.869	0.840	−2.304	4.997	1.810	0.189	0.558
	*p*-value	<0.001 ^t^	0.386 ^t^	0.402 ^t^	0.022 ^t^	<0.001 ^w^	0.072 ^t^	0.850 ^t^	0.577 ^t^
	Cohen’s d	0.83	0.01	−0.13	−0.45	1.08	0.05	0.02	0.10
Mania vs. Depression	*t*-value	1.135	1.227	1.109	2.748	−1.314	0.588	0.811	−0.693
	*p*-value	0.278 ^t^	0.223 ^t^	0.270 ^t^	0.007 ^t^	0.192 ^t^	0.558 ^t^	0.420 ^t^	0.490 ^t^
	Cohen’s d	0.27	0.50	0.24	0.59	−0.26	0.50	0.13	−0.15

^w^: welch test; ^t^: *t*-test.

**Table 5 brainsci-12-01034-t005:** Inflammatory ratios: neutrophil-to-lymphocyte (NLR), monocyte-to-lymphocyte (MLR), platelet-to-lymphocyte (PLR) and systemic immune-inflammatory (SII) index in male and female patients with mania or depression and healthy controls. Values are presented as mean ± SD.

		Mean ± SD Counts (Cells × /μL)							
		Male				Female			
		NLR	MLR	PLR	SII Index	NLR	MLR	PLR	SII Index
Control		1.8 ± 0.8	0.3 ± 0.1	106.7 ± 37.2	425.9 ± 213.3	1.9 ± 0.9	0.2 ± 0.1	129.4 ± 48.3	492.8 ± 213.3
Mania		2.5 ± 1.1	0.3 ± 0.1	111.4 ± 52.3	612.9 ± 306.4	2.3 ± 1.1	0.3 ± 0.1	123.7 ± 54.4	582.1 ± 338.8
Depression		2.5 ± 1.1	0.3 ± 0.1	101.1 ± 37.4	522.4 ± 259.8	2.7 ± 1.2	0.3 ± 0.1	135.0 ± 58.0	722.2 ± 379.3
		Comparisons between groups							
Mania vs. Control	*t*-value	4.794	1.975	0.787	4.602	2.321	1.597	−0.694	1.659
	*p*-value	<0.001 ^w^	0.049 ^t^	0.432 ^t^	<0.001 ^w^	0.024 ^w^	0.116 ^t^	0.488 ^t^	0.103 ^t^
	Cohen’s d	0.79	0.20	0.11	0.078	0.42	1.00	−0.11	0.37
Depression vs. Control	*t*-value	4.073	1.333	−0.770	2.241	4.122	1.634	0.647	3.611
	*p*-value	<0.001 ^t^	0.184 ^t^	0.442 ^t^	0.026 ^t^	<0.001 ^w^	0.103 ^t^	0.518 ^t^	0.001 ^w^
	Cohen’s d	0.83	0.20	−0.15	0.44	0.85	1.00	0.11	0.93
Mania vs. Depression	*t*-value	0.153	0.084	0.966	1.404	1.655	−0.254	−0.918	−1.788
	*p*-value	0.878 ^t^	0.933 ^t^	0.336 ^t^	0.164 ^t^	0.102 ^t^	0.800 ^t^	0.361 ^t^	0.077 ^w^
	Cohen’s d	0.02	0.20	0.21	0.31	−0.35	−0.20	−0.20	−0.39

^w^: welch test; ^t^: *t*-test.

**Table 6 brainsci-12-01034-t006:** Prognostic accuracy of white blood cells, platelets and Inflammatory Ratios.

Prognostic Marker	Cut-Off	Sensitivity	Specificity	PPV	NPV	AUC	95% CI	Sig.
Neutrophil	4.38	66.1%	71.4%	50.4%	82.7%	0.731	0.685–0.777	<0.001
Monocyte	0.59	53.9%	64.8%	40.2%	76.1%	0.604	0.555–0.653	<0.001
Lymphocyte	3.15	17.2%	90.2%	43.7%	71.2%	0.507	0.455–0.560	0.782
Platelet	246.5	51.7%	53.3%	32.7%	71.5%	0.496	0.445–0.548	0.888
NLR	2.15	57.8%	75.3%	50.7%	80.2%	0.690	0.643–0.737	<0.001
MLR	0.27	50.6%	63.6%	37.9%	74.5%	0.582	0.533–0.632	0.001
PLR	180.13	12.2%	91.2%	37.9%	70.2%	0.478	0.427–0.530	0.412
SII index	516.33	54.4%	69.4%	43.9%	77.6%	0.652	0.604–0.701	<0.001

PPV: Positive predictive value; NPV: Negative predictive value; AUC: area under the curve; CI: Confidence interval.

## Data Availability

The datasets used and/or analyzed during the current study are available from the corresponding author on reasonable request.

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
