# Peer review of "Neutrophil-to-Lymphocyte, Monocyte-to-Lymphocyte, Platelet-to-Lymphocyte Ratio and Systemic Immune-Inflammatory Index in Different States of Bipolar Disorder"

_brainsci, 2022, doi:10.3390/brainsci12081034_

Round 1

Reviewer 1 Report

In this study, the authors investigated that NLR, MLR, and SII index were significantly higher in patients with BD than in healthy controls, which imply a higher grade of inflammation in BD patients. Some concerns and suggestions are listed as below:

It is not clear for readers how these patients were treated in this study? I wonder if any drugs may affect the levels of NLR, MLR, and SII index. Detailed baseline characteristics should be provided.

The authors said that 180 patients and 409 healthy controls were included. How many patients were screened at the beginning? Did you exclude any patients?

Disease course of these patients should be provided in the revised manuscript.

Why Student’s T-test was used in this study? Please double check.

Levels of CRP and IL-6 should also be provided.

It is not clear for readers if NLR, MLR, and SII index had any correlations with clinical features of bipolar disorder.

Author Response

We thank the reviewer for their constructive remarks. We have taken the comments on board to improve and clarify the manuscript. Please find below a detailed point-by-point response to all comments.

Reviewer 2 Report

In their manuscript, the authors compared NLR, MLR, PLR, SII index of patients with BD and healthy controls and observed elevation in NLR, MLR and SII in BD (both phases) vs controls. In addition, they have examined possible sex-related differences in these measures and observed some differences. Although the manuscript is not ground-breaking, this manuscript is nicely written and it is of interest to the readers of Brain Sciences.

Major comments: I strongly recommend the authors to report test statistic values (t-test/welch) for all the tests conducted, compare BD-manic and BD-depressive individuals on the biomarkers of interest and compare males and females on the biomarkers of interest.

Minor comments:

Lines 47-48:- “the levels of chemokines associated with inflammation were found to be

48 higher in BD patients than in healthy controls” – please clarify which ones

Lines 72-73: “The systemic immune inflammation index (SII) is an innovative and prognostic index,

73 based on a combination of platelet, neutrophil, and lymphocyte counts” – please clarify how these biomarkers are combined

Line 88: What is the upper age limit ?

Line 92: Please add time of the blood draw. If it is between 7.30 - 9.00 am on line 103, please clarify in line 103 that all blood samples (BD patients and controls) were taken within these hours.

Table 1 – please conduct appropriate statistical tests and add p values to check whether groups were similar or different in terms of sex and age.

Section 3.3. I strongly recommend the authors to run subgroup ROC analyses (mania and depression) and report in the supplementary materials.

Author Response

We thank the reviewer for their constructive remarks. We have taken the comments on board to improve and clarify the manuscript. Please see in the attachment a detailed point-by-point response to all comments

Round 2

Reviewer 1 Report

The authors have answered my concerns. However, changes should be made (or discuss it as a limitation) in the revised manuscript when the authors fail to provide any data to address the concerns.

Reviewer 2 Report

I would like to thank to authors for addressing the majority of my comments. 

I have a couple of minor suggestions:

- please report t test statistics (not just p-values), as it gives an indication as to how big the effects are

- I appreciate that the time blood drawn might not be known for patients. Given that this may have an impact on the results, I would recommend authors to include and discuss this point as a limitation. 
